# Drug Repositioning for HPV Clade-Specific Cervicouterine Cancer Using the OCTAD Pipeline

**DOI:** 10.3390/ijms262211238

**Published:** 2025-11-20

**Authors:** Joel Ruiz-Hernández, Guillermo de Anda-Jáuregui, Enrique Hernández-Lemus

**Affiliations:** 1División de Genómica Computacional, Instituto Nacional de Medicina Genómica, Mexico City 14610, Mexico; josejoelruizhernandez@gmail.com (J.R.-H.); gdeanda@inmegen.edu.mx (G.d.A.-J.); 2Programa de Maestría y Doctorado en Ciencias Bioquímicas, Circuito de Posgrado, Universidad Nacional Autónoma de México, Ciudad Universitaria, Mexico City 04510, Mexico; 3Investigadores por Mexico, Conahcyt, Mexico City 03940, Mexico

**Keywords:** cervical cancer, human papillomavirus, HPV clades, drug repurposing, transcriptomic signatures, OCTAD framework, systems biology, precision oncology

## Abstract

Cervical cancer remains a major global burden largely caused by persistent infection with high risk human papillomavirus (HPV). Biological differences between HPV clade A7 and HPV clade A9 may influence tumor programs and clinical outcomes. To propose pharmacological candidates for repositioning, we applied an expression-based drug repurposing approach using the OCTAD (Open Cancer Therapeutic Discovery) framework. Disease transcriptional signatures were constructed for both HPV clades and compared with drug perturbation profiles to identify compounds showing inverse associations with the tumor related expression patterns, restricting the analysis to Food and Drug Administration (FDA) approved agents. The screening identified 41 and 52 candidates for HPV clade A7 and HPV clade A9, respectively, and stronger transcriptomic reversal was associated with higher drug sensitivity in relevant cell lines. These candidates were enriched for pharmacologic classes such as histone deacetylase inhibitors, estrogen pathway modulators, and statins. Additional enriched categories also emerged, including antimetabolites, protein kinase inhibitors, proteasome inhibitors, antimalarials, and antimicrobial agents, several of which already show experimental activity in cervical cancer models. These findings reveal both shared and clade-associated vulnerabilities in HPV-driven cervical cancer and demonstrate the utility of expression-based repurposing for generating actionable hypotheses. The resulting drug lists provide a concise, biologically grounded resource to guide preclinical validation and rational exploration in cervical cancer HPV positive models.

## 1. Introduction

Cervical cancer (CC) is a major public health concern and ranks fourth worldwide in both incidence and mortality among women [1]. A substantial share of the burden is concentrated in low and middle-income countries, where access to timely diagnosis and effective treatment remains limited and coverage of prophylactic vaccination programs is insufficient [2]. As a consequence, nearly 482,000 deaths are projected by 2040, representing a 38.2% increase compared with 2022 [1]. These figures underscore the urgent need to advance effective and accessible therapeutic strategies to improve survival and quality of life for patients.

Etiologically, CC is attributable almost entirely to persistent infection with high risk human papillomavirus (HPV) [3], whose oncogenes function as key carcinogenic drivers [4,5]. During tumor progression, HPV helps sustain malignancy and may promote invasiveness [6,7]. Interestingly, high-risk genotypes display marked biological diversity that shapes molecular programs and clinical outcomes [8]. This diversity is evident at the level of HPV phylogenetic clades, of which the most clinically relevant are alpha-7 (A7; comprising HPV 18, 39, 45, 59, 68, 70, 85, and 97 genotypes) and alpha-9 (A9; comprising HPV 16, 31, 33, 35, 52, 58, and 67 genotypes) [9,10]. In line with these groupings, tumors associated with HPV clade A7 are typically less differentiated, more likely to harbor integrated viral transcripts, and more aggressive, patterns that correlate with poorer prognosis and shorter recurrence-free survival [9,11]. By contrast, tumors linked to HPV clade A9, generally retain epithelial differentiation, show higher keratin expression, and predominantly exhibit episomal viral transcripts with lower levels of genomic integration, features that together are associated with comparatively better outcomes [9,11]. Despite these differences, it remains uncertain whether HPV genotype influences therapeutic response, because the available evidence is inconsistent [12,13,14]. Consequently, genotype is not incorporated into routine clinical decision making, and management of advanced CC continues to rely on surgery, radiotherapy, and chemotherapy with cisplatin as the mainstay despite substantial toxicity and limited specificity [15,16,17,18]. Although targeted agents directed at VEGF, EGFR, or PD-L1 have shown promise, their clinical use remains limited and overall prognosis is still poor [19,20].

Within this context, the search for new therapeutic strategies is constrained by the high cost, long timelines, and high attrition of de novo drug development. Only about one in 5000 to 10,000 anticancer candidates ultimately secures Food and Drug Administration (FDA) approval, and each new molecular entity typically requires more than US $2 billion and approximately 13 years from discovery to market [21]. As a practical alternative, drug repurposing identifies new indications for already approved agents and thereby offers a faster, less expensive, lower-risk, and operationally efficient pathway to effective treatments compared to novo discovery [22]. In oncology, multiple successes have been reported for drugs originating both within and outside cancer therapeutics [23]. However, progress in cervical cancer has been comparatively slow, largely due to tumor heterogeneity and the limited translation of preclinical findings into clinical benefit [20,24,25]. This highlights the need for systematic and rigorous evaluation of repositioning candidates to accelerate therapeutic advances in this disease.

Building on this landscape, drug repurposing has evolved from an empirical practice into a systematic and precision oriented discipline that integrates multi-omics and pharmacological resources with computational analytics to align disease mechanisms with existing agents, thereby advancing system level target discovery and therapeutic design [26,27,28]. A central pillar of this framework is expression-based connectivity mapping, which compares disease transcriptional profiles with drug perturbation signatures and prioritizes compounds whose profiles are inversely correlated with the disease state; this approach is grounded in the original Connectivity Map and its next generation LINCS compendium that together provide millions of drug–gene expression relationships for systematic inference [29,30]. Within oncology, methodologic syntheses and case studies show how primary in silico integration, secondary prioritization, and tertiary experimental validation can be assembled into an efficient pipeline for precision repurposing across tumor contexts [22,31,32]. Empirical applications of these strategies have demonstrated predictive accuracy and translational potential, with reversal drug signatures successfully applied and validated across multiple cancer contexts, including hepatocellular carcinoma, Ewing sarcoma, small cell lung cancer, glioblastoma, among others [33,34,35,36]. Although transcriptome-guided drug prioritization has also been applied to cervical cancer, available studies remain limited and methodologically diverse [37,38,39]. Expanding these efforts with systematic analyses that integrate patient-level transcriptional data and HPV clade-associated molecular programs may help refine candidate discovery for this malignancy.

In this context, the Open Cancer Therapeutic Discovery (OCTAD) platform provides an open framework for implementing transcriptome-based connectivity mapping across patient cohorts [33,40]. By constructing disease signatures, matching them against large drug perturbation repositories, and summarizing anticorrelation scores, the framework enables reproducible and scalable identification of compounds with potential activity across cancer subtypes [40]. Therefore, by leveraging transcriptional programs stratified by HPV clade and applying OCTAD screening, the present study aims to generate a well-supported and prioritized list of pharmacological candidates for cervical cancer that can serve as a robust resource for future preclinical and experimental validation.

## 2. Results

### 2.1. Differential Expression Analysis Across HPV Clades Supports Both Shared and Clade Related Transcriptional Programs

Building on previous genomic studies reporting that high risk HPV phylogenetic clades differ in molecular landscapes and clinical behavior [9,11,41], we divided our dataset of primary cervical carcinomas into HPV clade A7 and HPV clade A9 according to tumor genotype (Appendix A) [10,42]. Before the comparative analyses, we verified that key clinical features were similarly distributed between clades; both FIGO stage and histological type showed no significant differences (Appendix A). We then performed differential gene expression analysis for tumors versus controls using edgeR, DESeq2, and limma-voom (Figure 1A). Most genes showed consistent expression patterns across methods, and we defined a consensus signature as the intersection of the three analytical outputs. Based on this definition, we identified 1455 underexpressed and 548 overexpressed genes in HPV clade A7 tumors, while HPV clade A9 tumors displayed 2037 underexpressed and 883 overexpressed genes (Figure 1B). Interestingly, we observed overlap between the two clades, with 53.5% of overexpressed and 60.6% of underexpressed genes shared (Appendix A).

To explore the biological implications of these signatures, we performed functional enrichment analysis (Figure 1C). As expected, several pathways were shared between both clades, which is consistent with the fact that all samples correspond to primary tumors with comparable clinical stage and histological type distributions. These common processes primarily involved cell cycle regulation, chromosome segregation, extracellular matrix organization, and immune related functions. Beyond these similarities, each clade displayed distinct enrichment patterns: HPV clade A7 tumors displayed upregulation of cytoskeletal programs, including microtubule binding and annealing activity. In contrast, HPV clade A9 tumors exhibited upregulation of interferon mediated signaling and DNA helicase activity, alongside downregulation of histone kinase activity and intracellularly calcium gated channel activity. Overall, these results support the idea that, while tumors with presence of HPV clade A7 and HPV clade A9 share a broad transcriptional program, each one may exhibit distinct functional features consistent with their phylogenetic origin [11,41,43,44]. Such divergence has been suggested to contribute to differences in tumor biology and could potentially influence variability in treatment response [12,13,14].

### 2.2. Drug Candidates for Repositioning Across HPV Clade Groups in Cervical Cancer

Reversal based transcriptomic approaches have been used to identify compounds with expression profiles opposite to disease programs, showing consistent results across arious cancers and experimental models [33,45,46,47]. Following this rationale, we calculated summarized Reversal Gene Expression Scores (sRGES) values for drugs in HPV driven cervical cancer, thereby generating a comprehensive landscape of potential repositioning opportunities across both HPV clade A7 and HPV clade A9. For the HPV clade A7 signature, 5569 compounds were evaluated, of which 931 corresponded to FDA-approved drugs. In the case of HPV clade A9, we obtained 4014 compounds that were tested, including 693 FDA-approved.

To further contextualize these results, we performed enrichment analysis of the FDA-approved drugs (Appendix A). Using Medical Subject Headings categories (MeSH), we observed significant enrichment for pharmacological classes such as histone deacetylase (HDAC) inhibitors, selective estrogen receptor modulators (SERMs), estrogen antagonists, and Hydroxymethylglutaryl-CoA (HMG-CoA) reductase inhibitors, reflecting both well-established anticancer agents and broader therapeutic categories. In parallel, ChEMBL target-based annotations highlighted recurrent involvement of molecular families including HDAC isoforms, ATP-binding cassette (ABC) transporters, and nuclear hormone receptors. Notably, these enrichments were supported by multiple independent drugs converging on the same categories or targets, thereby reinforcing the biological plausibility and therapeutic relevance of the predicted candidates.

According to the OCTAD framework, drugs with summarized reversal gene expression scores (sRGES) below −0.20 are considered to exhibit the most representative and biologically meaningful reversal effects across benchmark analyses [40]. Following this criterion, we focused on the most significant candidates, defined as drugs with sRGES ≤ −0.20, and we identified 63 compounds that represent the strongest predicted reversal of the clade related transcriptional signatures (Figure 2). Of these, 31 were shared between HPV clade A7 and clade A9; however, their sRGES values varied across clades, suggesting that the same drug may exert a different predicted strength of transcriptomic reversal depending on the underlying clade-related expression profile. By contrast, the remaining 32 compounds were exclusive to a single group, defining clade-related candidate sets. Within the FDA-approved subset, 10 drugs met the significance threshold for HPV clade A7 and 21 for HPV clade A9. Together, these findings outline a therapeutic landscape that encompasses both broadly shared and clade restricted agents, emphasizing the potential to explore pan-clade repositioning strategies alongside approaches tailored to the distinct molecular contexts of HPV A7 and A9 tumors.

Heatmap of FDA-approved drug candidates with significant reversal potential (sRGES ≤ −0.20) across HPV clade A7 and HPV clade A9 cervical cancer signatures. Rows correspond to clades and columns to individual drugs, ordered by consensus sRGES. The color scale represents sRGES (darker blue = stronger predicted reversal). The annotation bar at the top indicates compounds common to both clades (green), unique to HPV clade A7 (orange), or unique to HPV clade A9 (purple).

### 2.3. Validation of Predicted Drug Response

Following the OCTAD workflow, we evaluated whether more negative sRGES scores translate into better pharmacologic response in the most relevant cell lines (Figure 3). Interestingly, we observed consistent and directionally concordant associations for both sample groups. In HPV clade A7, sRGES showed a positive correlation with the logarithm of the median half maximal inhibitory concentration (log10 [median IC50]) and a negative correlation with the area under the drug response curve (AUC). Here, IC50 represents the concentration of a drug required to reduce cell viability by 50%, with lower values indicating greater potency. On the other hand, the recomputed AUC, as defined in PharmacoGx, reflects the integral of the drug response across doses, calculated as the area of (1–viability); in this framework, higher AUC values indicate stronger overall sensitivity (Appendix A). Thus, our results show that stronger transcriptomic reversal, captured by more negative sRGES values, is associated simultaneously with lower IC50 and higher AUC. The same trend was observed in HPV clade A9, supporting that negative sRGES values consistently track with increased pharmacologic sensitivity across clades. Together, the concordance between sRGES and drug response strengthens the evidence that our top candidates provide a robust foundation for experimental validation and clade aware drug repositioning.

### 2.4. Evidence in Cervical Cancer Models for Predicted Candidates

Using the PubMed strategy described in Section 4.7, we conducted a targeted literature screen covering recent studies to identify previous experimental or clinical evidence supporting our predicted drug candidates in cervical cancer (Appendix A). This analysis revealed compounds that have reached clinical evaluation, including vorinostat and SN-38 as active components, which have been tested in phase II or advanced clinical settings in recurrent or metastatic cervical cancer [48,49,50], as well as etoposide in combination regimens [51]. Several additional candidates showed antitumor activity in vivo in cervical cancer models, including palbociclib, fulvestrant, pitavastatin, simvastatin, methotrexate, pyrvinium, and closantel, through effects such as tumor growth suppression, radiosensitization, or chemotherapy potentiation [52,53,54,55,56,57]. A broader set of drugs, such as ivermectin, amodiaquine, quinine, bortezomib, dasatinib, raloxifene, duloxetine, panobinostat, nifuroxazide, and penfluridol, demonstrated in vitro activity in cervical cancer cell lines, including cytotoxicity, apoptosis induction, pathway inhibition, and enhanced response to standard agents [58,59,60,61,62,63,64,65,66,67,68]. The complete list of compounds is provided in Appendix A, which offers a qualitative, study level corroboration of our computational predictions.

## 3. Discussion

Accumulating evidence indicates that phylogenetic clades of high risk HPV contribute to distinct biological programs in CC [11,41], and our results are consistent with this knowledge. The clade stratified transcriptional signatures in this study revealed a shared malignant core between HPV clade A7 and HPV clade A9, encompassing pathways linked to known drivers of cervical carcinogenesis such as epigenetic dysregulation and histone deacetylase activity [69,70]. At the same time, we observed clade-related differences in gene expression patterns. These differences appear biologically consistent with prior multi-omics studies reporting clade dependent epigenetic modulation, including variation in DNA methylation, chromatin organization, and viral–host interactions [11]. In this context, there are several plausible contributors to this divergence, one of which could be the behavior of the viral genome: HPV18 (HPV clade A7) is frequently found fully integrated into the host genome, whereas HPV16 (HPV clade A9) can persist partially as episomal DNA and shows a lower integration frequency [41,71]. Integration events often occur at fragile or regulatory genomic regions and can generate highly active enhancer regions that amplify nearby oncogenes and reshape local transcriptional programs [71,72].

However, several aspects should be considered when interpreting these findings. It is well established that CC biology is shaped by both intrinsic host factors and viral characteristics [73]. In this context, although we did not detect major differences in key clinical features between the analyzed HPV clade groups, the transcriptional distinctions observed here may also reflect additional layers of heterogeneity, including patient-specific biology, features of the tumor microenvironment, and immune composition [74,75]. Moreover, each HPV clade encompasses multiple genotypes, which may introduce variability not fully captured in our analysis. Nevertheless, classifying CC according to HPV lineage provides a more refined alternative to treating all tumors as a single group and offers a biologically grounded rationale for therapeutic decision making. Since HPV genotyping is already part of routine clinical practice [15], clade-based stratification may represent a practical and clinically meaningful extension that could inform future treatment prioritization.

This study also has data-level constraints. All disease signatures were derived from bulk RNA-seq, which captures an averaged signal across tumor, immune, stromal, and vascular compartments [76]. Consequently, the observed clade-related transcriptomic differences may also be shaped by variation in cellular composition and the tumor microenvironment [77]. To enhance the biological resolution of future analyses, single-cell and spatial transcriptomic data should be integrated to separate epithelial from non-epithelial components, elucidate how HPV lineage influences tumor and microenvironmental programs, and evaluate drug reversal within specific cellular contexts [78]. In addition, this study relies entirely on publicly available datasets, which ensure reproducibility and transparency but inherently limit discovery to the scope of existing cohorts and perturbational profiles. Integrating these large-scale resources with newly generated experimental data would allow the validation and refinement of the computational predictions presented here. Despite these limitations, bulk transcriptomic reversal based on public data remains a powerful and predictive strategy, as similar frameworks have successfully identified active compounds that later advanced to translational validation [33,34,35,36], underscoring its continued relevance as a robust approach for uncovering biologically meaningful therapeutic candidates.

Methodologically, we used the OCTAD framework to prioritize compounds whose perturbational transcriptomic profiles oppose the clade-related disease signatures in CC. We adopted thresholds for cell line similarity and transcriptomic reversal scoring (sRGES) that were established and validated by OCTAD through extensive benchmarking across pharmacogenomic datasets, which demonstrated consistent associations between reversal strength and drug sensitivity [40,79]. Nevertheless, these thresholds are not absolute, and modifying them may influence the number of prioritized compounds. In this context, it would be of great interest for future studies to explore alternative parameter settings; however, assessing whether such modifications could yield improved predictive or therapeutic performance would require systematic experimental benchmarking. Even so, our analysis supported the validity of these parameters, as the obtained sRGES values showed coherent correlations with drug sensitivity across representative cell lines and aligned with recent reports, since some of the top-ranked compounds have already shown experimental or clinical activity in CC. Although we focused on the top predicted candidates in this study, we also provide the complete list of evaluated drugs to ensure transparency and reproducibility. Altogether, these findings support the robustness of our results and position this work as a biologically grounded approach that generates hypotheses for drug repositioning in HPV-related CC.

These correlations supported the internal consistency of our results but should be interpreted with caution, since pharmacogenomic cell lines may not fully recapitulate HPV clade–specific diversity. Future work could strengthen this validation through regression models that account for additional biological factors. Nevertheless, our literature assessment revealed that several of the prioritized compounds already exhibit antitumour activity in cervical cancer, whereas others remain unexplored and represent promising opportunities for future research. To strengthen these findings, upcoming studies should explore the top-ranked compounds in models representative of both HPV clades, assessing viability, apoptosis, and pathway modulation such as HDAC inhibition or estrogen receptor signaling. Complementary in vivo approaches using xenograft systems would provide stronger biological evidence and enhance the translational relevance of these predictions. Altogether, such studies could yield direct mechanistic insight and refine the therapeutic potential of the proposed compounds.

Building on these considerations, the reversal analysis revealed coherent pharmacological themes that mirror dysregulated biology in HPV-associated cervical cancer. Histone deacetylase inhibition emerged across clades, consistent with epigenetic alterations [80]. HDACs regulate chromatin structure and transcription, and their abnormal activity silences tumor-suppressor genes while sustaining the functions of viral oncoproteins [72,81]. Inhibition of these enzymes restores pro-apoptotic and cell-cycle control, as demonstrated for vorinostat through blockade of HPV18 DNA amplification and for panobinostat through induction of apoptosis via autophagy [63,82,83]. Early clinical exploration has also shown encouraging results, including the combination of the HDAC inhibitor vorinostat with the immune checkpoint inhibitor pembrolizumab in recurrent cervical and other squamous carcinomas, and increased immune activation when HDAC inhibition is combined with proteasome blockade [48,83].

We also identified hormone-modulating agents among the top-ranked candidates, consistent with the established role of estrogen signaling in HPV-related CC. In HPV transgenic models, fulvestrant and raloxifene induced pronounced tumor regression [56,84], while in cervical cell systems, raloxifene suppressed oestradiol-driven proliferation and reduced E6/E7 transcription [67], supporting the biological plausibility of anti-estrogen strategies in HPV-positive CC.

On the metabolic front, mevalonate-pathway blockade attenuates oncogenic signaling by reducing prenylation of small GTPases, thereby impacting RAS–MAPK and PI3K–AKT cascades [85]. In CC models, simvastatin suppressed proliferation, induced apoptosis, and potentiated cytotoxic chemotherapy via broad inhibition of prenylation-dependent pathways [57], while pitavastatin promoted oxidative-stress-mediated DNA damage with caspase-9/3 activation and PARP cleavage [86,87], highlighting metabolic rewiring as a shared vulnerability and a rationale for combinatorial testing.

Signals also appeared among antimicrobial/antiparasitic agents. Pyrvinium inhibited Wnt/β-catenin, triggered apoptosis, and potentiated cisplatin in CC systems [52]; closantel reduced angiogenesis and tumor growth in zebrafish xenografts [53]; monensin curtailed tumor growth in vivo while targeting Wnt/β-catenin [88]; and nifuroxazide suppressed STAT3 with pro-apoptotic effects [61]. By contrast, ivermectin showed antiproliferative effects only at supraphysiological concentrations, limiting its translational feasibility [58,89]. These drug classes should therefore undergo careful evaluation of potency and therapeutic relevance in HPV-associated CC before any clinical consideration.

Several recovered agents overlapped with conventional cytotoxics. DNA-targeting drugs likely reflect broad antitumour activity; nonetheless, irinotecan (SN-38) combined with cisplatin has demonstrated efficacy in CC, with phase II response rates approaching 60% [49,50,90], while etoposide plus cisplatin has provided consistent benefit in recurrent settings [51,91,92]. Mechanistically compelling candidates also included bortezomib, which stabilizes p53 and PDZ-domain tumor suppressors (hDlg, hScrib) to counteract HPV oncoproteins [64,65], producing synergistic tumor regression when combined with cisplatin in HeLa xenografts [93].

Additional targeted signals were noted for CDK4/6 and mTOR pathways: palbociclib limited proliferation and induced apoptosis via oxidative stress in CC models, and in combination with SHetA2 suppressed phospho-Rb and tumor growth in SiHa xenografts [55,94]. By contrast, temsirolimus achieved disease stabilization with few objective responses in recurrent/metastatic disease, underscoring the need for biomarker-guided or combinatorial approaches [95]. We also observed activity from antipsychotic agents modulating calmodulin, PI3K/AKT/mTOR, and autophagy (penfluridol, chlorpromazine), although dose requirements and CNS adverse effects pose translational constraints [62,96].

Many of the pharmacologic classes prioritized by our signatures are mechanistically consistent with CC biology, reflecting a convergence that strengthens confidence in shared vulnerabilities such as epigenetic reactivation and refines hypotheses about lineage-related mechanisms. Although some of these agents have shown variable efficacy in the literature, others remain untested in CC models, offering new opportunities for exploration. Our results narrow the landscape to FDA-approved compounds with well-characterized safety, pharmacokinetic, and clinical profiles, which facilitates translational progress [22]. Nevertheless, evaluating predicted agents that are not yet FDA-approved could uncover additional mechanisms or therapeutic opportunities beyond current pharmacological boundaries. Building on this foundation, rational combination strategies, for example, pairing HDAC inhibition with mevalonate pathway blockade or ER antagonism with proteasome inhibition, represent a logical next step, though their implementation must address the practical challenges of polypharmacology, including drug–drug interactions, overlapping toxicities, and regimen optimization [27]. Altogether, our prioritized compounds and their underlying mechanisms provide a coherent framework that bridges computational discovery with future experimental strategies in cervical cancer models, offering a foundation for biologically grounded exploration in HPV-associated CC.

## 4. Materials and Methods

### 4.1. Data and Cohort Definition

All analyses were conducted in R (v4.5.1) using octad (v1.9.0) and octad.db (v1.10.0) packages [40,79]. RNA-seq expression profiles were obtained from the OCTAD platform, which integrates uniformly Toil-processed RNA-seq data (HDF5 format) from TCGA and GTEx and the associated sample metadata (dataset EH7274), allowing for reliable cross-cohort comparisons (Figure 4). From the TCGA-CESC project, we selected 267 cervical carcinoma samples. HPV genotypes were assigned according to the annotations reported with the HPV-EM algorithm [42], and only mono-infected cases with high-risk types were considered. Therefore, samples were stratified into two phylogenetic clades: HPV clade A7 (*n* = 65; HPV 18, 39, 45, 59) and HPV clade A9 (*n* = 202; HPV 16, 31, 33, 35, 52, 58) [9]. The control group consisted of 13 normal cervical tissue samples, including 10 from GTEx and 3 adjacent normals from TCGA.

To evaluate the clinical comparability between HPV clades prior to transcriptomic analyses, contingency tables were constructed for FIGO stage and histological type [97,98,99], and differences in distribution were assessed using Fisher’s exact test implemented in R.

### 4.2. Differential Expression Analysis

Differential gene expression between tumors and controls was assessed separately for each clade group (HPV clade A7 and HPV clade A9) using three methods: edgeR, DESeq2, and limma-voom. In order to mitigate unwanted variation, RUVSeq was applied with *k* = 1 for edgeR and DESeq2, following OCTAD’s default pipeline. Genes were considered significantly differentially expressed when |log_2_FC| ≥ 1 and FDR ≤ 0.01, based on the adjusted *p*-values returned by each method.

### 4.3. Over-Representation Analysis

To explore the functional relevance of the consensus DEG signatures, we performed Over-Representation Analysis (ORA) using the clusterProfiler (v4.16.0) R package [100]. For each clade (HPV clade A7 and HPV clade A9), up- and downregulated genes were analyzed independently across the three gene ontology (GO) domains: biological process (BP), cellular component (CC), and molecular function (MF). ORA was conducted to capture all significant categories (*p* ≤ 0.05, q ≤ 0.20, BH-adjusted), and subsequently refined by collapsing redundant terms based on semantic similarity (cutoff = 0.25), yielding a simplified set of representative GO categories for each clade and regulation group.

### 4.4. sRGES Calculation

Candidate drugs were prioritized using summarized Reversal Gene Expression Scores (sRGES) implemented in the OCTAD framework [1]. This metric integrates differential expression (DE) signatures with cohort–cell line similarity weights to estimate the ability of a compound to reverse disease-associated transcriptional profiles. Similarity between tumor cohorts (HPV clade A7 and HPV clade A9) and cancer cell lines was first assessed with the function computeCellLine using the octad.small reference set; only lines with a median correlation (medcor) > 0.30 were retained. Clade-related DE signatures were then analyzed with the function runRGES, which assigns each compound an individual RGES defined as the difference between enrichment scores of up- and downregulated disease genes within LINCS L1000 perturbation profiles. The resulting scores were adjusted by the retained cell-line weights and summarized into an sRGES per drug [33].

Within each group of samples, only drugs consistently identified across the three DE methods were considered, with a consensus sRGES calculated as the arithmetic mean of the method-specific values. Analyses were then restricted to FDA-approved compounds, and the final lists were filtered to retain only candidates with consensus sRGES ≤ −0.20. All clade-related predictions were archived for reproducibility (Appendix A).

### 4.5. Drug Set Enrichment Analysis

To further contextualize the drug predictions, we carried out enrichment analysis of the consensus FDA-approved sRGES lists using the octadDrugEnrichment function. Two complementary classifications were examined: (i) MeSH (Medical Subject Headings), which organizes compounds by pharmacological action or therapeutic application, and (ii) ChEMBL targets, which group drugs according to annotated protein targets or families. Enrichment significance was evaluated at FDR ≤ 0.01, and for each enriched set, we additionally reported the number of predicted compounds contributing to the signal.

### 4.6. In Silico Validation of Predicted Drug Response

To evaluate the robustness of the predictions, we validated the consensus sRGES values against pharmacogenomic response data from cancer cell lines. For each group of samples (HPV clade A7 and HPV clade A9), we applied the OCTAD function topLineEval, which compares drug-specific sRGES scores with experimental sensitivity profiles available for the corresponding cell lines. The function returns two complementary measures for all overlapping compounds: (i) the area under the drug response curve (AUC), as recomputed in PharmacoGx from raw viability data and defined as the integral of (1–viability) across the tested dose range (higher AUC indicates greater sensitivity), and (ii) the median inhibitory concentration (IC_50_), reported after log_10_ transformation, representing the concentration required to reduce viability by 50% (lower IC_50_ indicates higher potency). Pearson’s correlation coefficients (r) and two-sided *p*-values were computed and adjusted using the Bonferroni method, with *p* < 0.01 considered statistically significant.

### 4.7. Literature Evaluation of Repositioned Drug Candidates

To identify which of the proposed drug candidates had already been experimentally or clinically evaluated in the context of cervical cancer, we performed a targeted literature search in PubMed. To ensure transparency and reproducibility, the search strategy used Boolean operators combining each compound’s International Nonproprietary Name (INN) with cervical cancer–related terms. A representative query was: (cancer AND (cervix OR cervical OR cervicouterine)) AND (“1 January 2015” [Date—Publication]:“1 October 2025” [Date—Publication]) AND (“<drug INN>”)). The search covered studies published between January 2015 and September 2025. Retrieved records were screened and classified according to the type of experimental validation: in vitro (cervical cancer–derived cell lines), in vivo (animal or xenograft models), or clinical (studies involving human subjects). Only compounds supported by at least one experimentally validated or clinical report in cervical carcinoma were retained.

## 5. Conclusions

By integrating CC transcriptional profiles with the OCTAD framework, we identified 41- and 52 FDA-approved compounds associated with the HPV A7 and HPV A9 clades, respectively. These compounds showed consistent associations between predicted transcriptomic reversal and drug sensitivity. Several of the prioritized agents already have experimental or clinical support in CC, while others represent new opportunities emerging from this hypothesis-generating study. Altogether, the resulting drug lists provide a concise and biologically grounded foundation to guide future experimental research and rational combination strategies in HPV-positive cervical cancer.

## Figures and Tables

**Figure 1 ijms-26-11238-f001:**
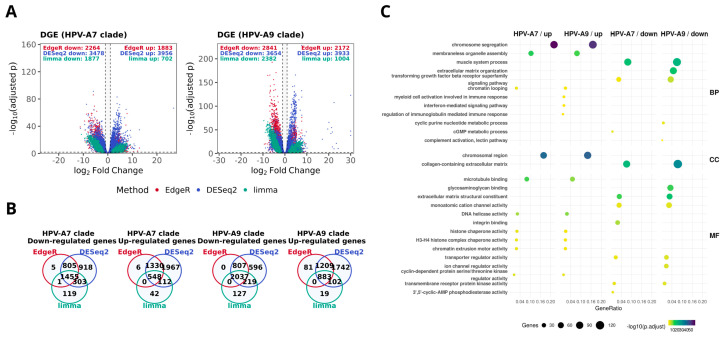
Differential expression signatures in HPV-associated cervical cancer by clade. (**A**) Volcano plots show significantly overexpressed and underexpressed genes in tumors with presence of HPV clade A7 and HPV clade A9, computed with edgeR (red), DESeq2 (blue), and limma-voom (green). The dotted lines indicate the thresholds used to define differential expression (|log_2_FC| ≥ 1 and FDR ≤ 0.01). Numbers of differentially expressed genes (DEGs) are indicated for each method. (**B**) Venn diagrams depicting the overlap of overexpressed and underexpressed DEGs across methods within each clade. (**C**) Gene Ontology (GO) enrichment analysis of consensus signatures. Selected biological processes (BP), cellular components (CC), and molecular functions (MF) are displayed for over- and underexpressed genes in each clade. Dot size reflects the number of genes per category, and color represents enrichment significance (−log10 adjusted *p*-value).

**Figure 2 ijms-26-11238-f002:**
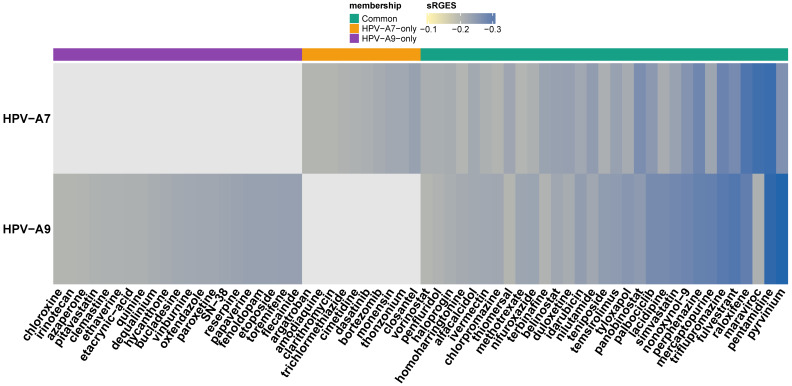
Drug candidates for repositioning in α-A7 and α-A9 cervical carcinomas.

**Figure 3 ijms-26-11238-f003:**
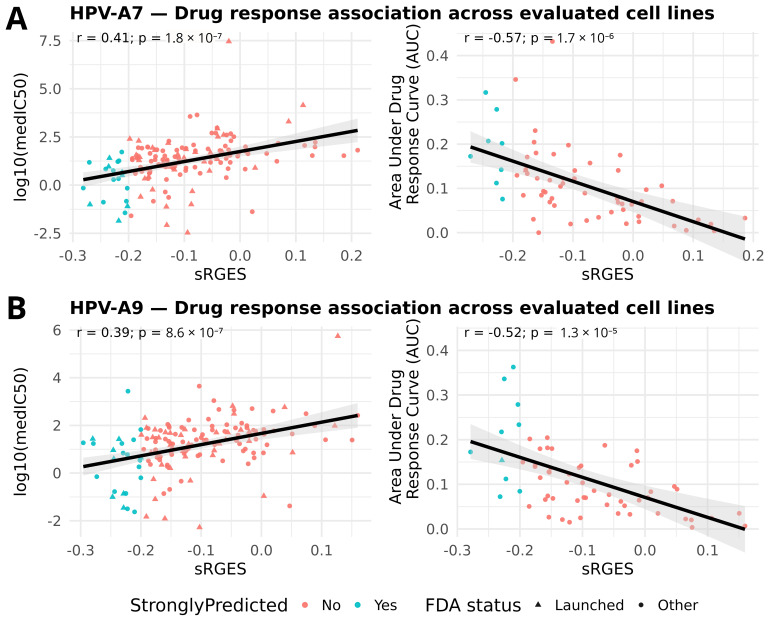
Validation of sRGES with pharmacologic response. (**A**) HPV clade A7; (**B**) HPV clade A9. Scatterplots show the relationship between sRGES (x-axis) and log_10_(median IC_50_) (left) or AUC (recomputed) (right), each point representing a candidate compound (median across tested lines). The black line shows the least-squares regression; the grey area represents its 95% confidence band.; in-panel labels display Pearson correlation (r) and nominal *p*-values. All tests remained significant after Bonferroni correction (A7 IC_50_: p_adj = 7.2 × 10^−7^; A7 AUC: p_adj = 6.8 × 10^−6^; A9 IC_50_: p_adj = 3.4 × 10^−6^; A9 AUC: p_adj = 5.2 × 10^−5^). Points are colored by prediction strength (sRGES ≤ −0.20) and shaped by FDA status (triangle = FDA-launched; circle = other).

**Figure 4 ijms-26-11238-f004:**
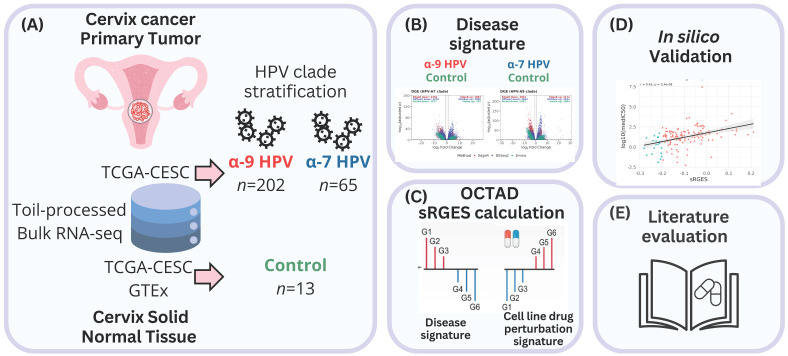
Graphical overview of the study design and analytical workflow. (**A**) Bulk RNA-seq data uniformly processed by Toil were obtained from TCGA-CESC tumors and GTEx/TCGA normal cervix samples. HPV genotypes were assigned with HPV-EM, and only mono-infected cases were retained. Tumor samples were stratified into HPV clade A9 (*n* = 202) and HPV clade A7 (*n* = 65) clades, with controls from normal cervix tissues (*n* = 13). (**B**) Differential expression analysis was performed separately for each clade versus controls using edgeR, DESeq2, and limma-voom with RUVSeq correction. Consensus DEG signatures were functionally interpreted by GO over-representation analysis. (**C**) Drug predictions were derived through OCTAD by calculating summarized Reversal Gene Expression Scores (sRGES), integrating tumor–cell line similarity and LINCS L1000 perturbation profiles, and filtered for FDA-approved compounds with sRGES ≤ −0.20. (**D**) Finally, predictions were validated in silico against pharmacogenomic sensitivity data (AUC and IC_50_) from cancer-relevant cell lines. (**E**) A literature search was performed in PubMed to identify which predicted drug candidates had prior experimental or clinical evidence in cervical cancer.

## Data Availability

All code involved in this work is publicly available at https://github.com/JoelRuiz26/Octad_Cervical_Cancer.git (accessed on 28 October 2025).

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
