# Peer review of "Drug Repositioning for HPV Clade-Specific Cervicouterine Cancer Using the OCTAD Pipeline"

_ijms, 2025, doi:10.3390/ijms262211238_

Round 1

Reviewer 1 Report

Comments and Suggestions for Authors

Dear authors,

The paper presents an interesting approach to drug repurposing for HPV-related cervical cancer, leveraging clade-specific transcriptomic signatures. The use of the OCTAD pipeline is well-justified, and the identification of potential drug candidates is promising. However, there are several areas where the manuscript could be strengthened to increase its rigor and impact.

  1. Limited Biological Validation - the “in silico” validation using cell line data is a good starting point, but it relies on existing pharmacogenomic datasets. The cell lines used might not perfectly represent the “in vivo” tumor microenvironment or the full spectrum of HPV-related cervical cancer subtypes. Stronger evidence would come from “in vitro” experiments specifically designed to test the top drug candidates in cell lines representing both HPV-A7 and HPV-A9 cancers. Ideally, these experiments should include assays for cell viability, apoptosis, and relevant pathway modulation (e.g., HDAC inhibition, estrogen receptor signaling). I suggest, even better, in vivo studies in xenograft models would add considerable confidence to the data.
  1. Confounding Factors and Collinearity - the authors acknowledge that "bulk RNA-seq signatures conflate epithelial and microenvironmental signals; HPV genotype, histology, and integration status are partially collinear." Well… the manuscript doesn't fully address the potential impact of these confounding factors. Please, a more detailed discussion of the limitations of using bulk RNA-seq is needed. Single-cell RNA-seq would provide better resolution of the cellular composition and cell-specific responses to drugs. The authors should also consider incorporating more detailed clinical data (e.g., stage, grade, treatment history) in their analysis to address the influence of these factors on the results.
  1. sRGES Threshold and Prioritization - The choice of sRGES ≤ −0.20 as a threshold for significant reversal is somewhat arbitrary…!!! The sensitivity of the downstream in silico validation may be affected. A sensitivity analysis should be performed to justify the threshold choice. You should explain the selection criteria for selected drugs.
  2. Limited Novelty Regarding Drug Targets - While the study identifies several potential drug candidates, many of the enriched pharmacological classes (e.g., HDAC inhibitors, estrogen modulators) are already known to be relevant in cervical cancer. Discuss the potential for combination therapies based on the identified drug classes.
  3. The study relies heavily on publicly available datasets (TCGA, GTEx, LINCS). While this is a strength in terms of reproducibility, it also limits the potential for novel discoveries. Please acknowledge the limitation!
  4. The paper mentions that "most prior efforts have generated large drug catalogs without stratifying by viral genotype." However, it doesn't provide a direct comparison of the identified drug candidates with those from other drug repurposing studies in cervical cancer. A table comparing the top candidates from this study with those from other studies would help to contextualize the findings and highlight the added value of the clade-specific approach.
  5. The validation of predicted drug response relies on correlation analysis. Authors should specify what adjustments they have made to deal with multiple testing issues, as well as justify the selection of the applied methods. Also, a more robust statistical approach could include regression modeling to predict drug response based on sRGES values, while controlling for other relevant factors.

Furthermore, the writing could be more concise and focused. Some sections contain excessive background information that could be shortened.

  • The figures are generally clear, but the axis labels in Figure 4 could be larger and more readable!
  • What criteria were used to select the cell lines for the “in silico” validation?
  • How sensitive are the results to the choice of RUVSeq normalization? Did the authors explore other normalization methods?
  • Can the authors provide more details on the potential mechanisms of action of the top drug candidates in the context of HPV-related cervical cancer?
  • What are the potential limitations of using FDA-approved drugs, given that some of these drugs may have significant side effects or limited bioavailability????

This study has the potential to make a valuable contribution to the field of drug repurposing for cervical cancer. However, the manuscript must be strengthened by addressing the abovementioned limitations, particularly by including more robust biological validation and providing a more nuanced discussion of the confounding factors. By addressing these concerns, the authors can increase the impact and credibility of their findings.

Author Response

Dear authors,

The paper presents an interesting approach to drug repurposing for HPV-related cervical cancer, leveraging clade-specific transcriptomic signatures. The use of the OCTAD pipeline is well-justified, and the identification of potential drug candidates is promising. However, there are several areas where the manuscript could be strengthened to increase its rigor and impact.

The authors are grateful to Reviewer 1 for the insightful comments and suggestions made about our work. In what follows we will present a point-by-point response to all of them. To ease reading our responses appear in bold type.

Limited Biological Validation - the “in silico” validation using cell line data is a good starting point, but it relies on existing pharmacogenomic datasets. The cell lines used might not perfectly represent the “in vivo” tumor microenvironment or the full spectrum of HPV-related cervical cancer subtypes. Stronger evidence would come from “in vitro” experiments specifically designed to test the top drug candidates in cell lines representing both HPV-A7 and HPV-A9 cancers. Ideally, these experiments should include assays for cell viability, apoptosis, and relevant pathway modulation (e.g., HDAC inhibition, estrogen receptor signaling). I suggest, even better, in vivo studies in xenograft models would add considerable confidence to the data.

We appreciate this thoughtful observation. In the revised version, we expanded the Discussion to acknowledge the scope of our current validation and to highlight the value of complementary experimental approaches. This suggestion is indeed valuable, as it provides a clear direction for future research to build on our findings. Accordingly, we added a paragraph emphasising that while pharmacogenomic datasets support the internal consistency of our predictions, further in vitro and in vivo studies in models representative of HPV-A7 and HPV-A9 tumours would be instrumental for deepening the biological understanding and advancing translational applications (see revised Discussion, lines 320–332).

Confounding Factors and Collinearity - the authors acknowledge that "bulk RNA-seq signatures conflate epithelial and microenvironmental signals; HPV genotype, histology, and integration status are partially collinear." Well… the manuscript doesn't fully address the potential impact of these confounding factors. Please, a more detailed discussion of the limitations of using bulk RNA-seq is needed. Single-cell RNA-seq would provide better resolution of the cellular composition and cell-specific responses to drugs. 

The authors should also consider incorporating more detailed clinical data (e.g., stage, grade, treatment history) in their analysis to address the influence of these factors on the results.

We thank the reviewer for this valuable comment. In the revised version, we expanded both the Methods and Discussion sections to provide a more comprehensive treatment of potential confounding factors and the intrinsic limitations of bulk RNA-seq data. In the Data and cohort definition section (lines 439–441), we now describe statistical comparisons between HPV clade samples using Fisher’s exact tests to assess potential differences in FIGO stage and histological type, confirming that the key clinical features were similarly distributed between clades. These results are also reported in the Results section (lines 119–124) and summarised in Supplementary Tables S1–S3. Furthermore, in the Discussion (lines 267–281), we expanded our analysis to acknowledge that several aspects should be considered when interpreting these findings: although the clinical characteristics appeared balanced, additional biological, viral genotype, and microenvironmental factors may additionally contribute to transcriptional variability across clade groups.

We also included a dedicated paragraph discussing the resolution limits of bulk RNA sequencing (lines 282–298) and emphasised that single cell and spatial transcriptomic approaches will be essential to achieve higher biological resolution. At the same time, we note that the bulk RNA-seq transcriptomic reversal approach remains a valuable exploratory approach with proven ability to capture biologically relevant signals.

sRGES Threshold and Prioritization - The choice of sRGES ≤ −0.20 as a threshold for significant reversal is somewhat arbitrary…!!! The sensitivity of the downstream in silico validation may be affected. A sensitivity analysis should be performed to justify the threshold choice. You should explain the selection criteria for selected drugs.

In the revised version, we clarified the rationale for selecting the sRGES threshold (≤ −0.20) within the empirically supported range established by OCTAD benchmarking. As noted in the Discussion (lines 300–318), this cutoff does not compromise the robustness of our in silico validation, since the analysis evaluates the full spectrum of sRGES values, consistently showing stronger drug sensitivity for more negative scores. We also emphasise that, although our main focus is on compounds below −0.20, the complete list of evaluated drugs, covering a broader sRGES range, is provided in the Supplementary Material to ensure transparency and reproducibility.

Limited Novelty Regarding Drug Targets - While the study identifies several potential drug candidates, many of the enriched pharmacological classes (e.g., HDAC inhibitors, estrogen modulators) are already known to be relevant in cervical cancer. Discuss the potential for combination therapies based on the identified drug classes.

In the revised Discussion (lines 402–410), we included a dedicated section addressing the potential for combination therapies derived from the prioritised drug classes. This paragraph highlights how the identified pharmacologic patterns can inform rational combination strategies and serve as a foundation for future preclinical exploration in HPV-associated cervical cancer.

The study relies heavily on publicly available datasets (TCGA, GTEx, LINCS). While this is a strength in terms of reproducibility, it also limits the potential for novel discoveries. Please acknowledge the limitation!

We thank the reviewer for this valuable observation. In the Discussion (lines 290–294), we briefly acknowledge this limitation and mention the implications of relying on publicly available datasets.

The paper mentions that "most prior efforts have generated large drug catalogs without stratifying by viral genotype." However, it doesn't provide a direct comparison of the identified drug candidates with those from other drug repurposing studies in cervical cancer. A table comparing the top candidates from this study with those from other studies would help to contextualize the findings and highlight the added value of the clade-specific approach.

We added a new Results section (2.4, lines 233–249) where we conducted a targeted literature review following the strategy described in the Methods (section 4.7). This section provides an updated summary of drug repurposing studies reporting experimental or clinical evidence supporting our prioritised compounds, as presented in Supplementary Table S4. Furthermore, we refer to this analysis in the Discussion (lines 324–326), where we note that several of the prioritised drugs already show antitumour activity in cervical cancer, while others remain unexplored and represent promising opportunities for future investigation.

The validation of predicted drug response relies on correlation analysis. Authors should specify what adjustments they have made to deal with multiple testing issues, as well as justify the selection of the applied methods. Also, a more robust statistical approach could include regression modeling to predict drug response based on sRGES values, while controlling for other relevant factors.

We thank the reviewer for this thoughtful comment. In the revised version, we clarified in the Methods section (4.6) that the validation analysis followed the OCTAD workflow, which computes Pearson correlations between predicted sRGES values and pharmacogenomic sensitivity metrics (AUC and ICâ‚…â‚€). The resulting p-values were adjusted for multiple testing using the Bonferroni correction, and values below 0.01 were considered statistically significant, as also indicated in the Figure 4 legend. As noted in the Discussion (lines 320–332), while this approach provides a consistent and transparent validation framework, future analyses could incorporate regression-based models to account for additional biological covariates and further refine drug–response predictions.

Furthermore, the writing could be more concise and focused. Some sections contain excessive background information that could be shortened.

In the revised manuscript, we reduced background content throughout the Introduction and Discussion, keeping only information directly relevant to the study’s objectives and findings to improve clarity and focus.

The figures are generally clear, but the axis labels in Figure 4 could be larger and more readable!

We thank the reviewer for this helpful suggestion. Figure 4 has been redrawn with larger axis labels and improved formatting to enhance readability.

What criteria were used to select the cell lines for the “in silico” validation?

As described in the Methods (section 4.6), cell lines were selected following the OCTAD framework, which defines representative models based on transcriptomic similarity. The rationale is that cell lines with similar transcriptomic signatures capture similar biology, thus making appropriate validation datasets.  This aspect and its implications are also discussed in the revised Discussion (lines 300–318).

How sensitive are the results to the choice of RUVSeq normalization? Did the authors explore other normalization methods?

We thank the reviewer for this pertinent question. In this study, normalization was performed using the RUVSeq implementation provided in the OCTAD workflow to ensure methodological consistency. We did not explore alternative normalization strategies, since this particular normalization approach is designed for this type of study. This is so since (unlike common normalization methods) is able to minimize the biases due to the combination of several data sources (large batch effects).

Can the authors provide more details on the potential mechanisms of action of the top drug candidates in the context of HPV-related cervical cancer?

The potential mechanisms of action of the top-ranked compounds are discussed throughout the Discussion section (lines 336–391), where we address the main pharmacologic classes and highlight both their established and plausible roles in cervical cancer biology.

What are the potential limitations of using FDA-approved drugs, given that some of these drugs may have significant side effects or limited bioavailability????

We acknowledge the limitation of focusing primarily on FDA-approved compounds (lines 398–402). While this approach may restrict the discovery of entirely novel agents, it offers the advantage that their safety, toxicity, and pharmacokinetic profiles are already well established, allowing dose adjustments to be explored within known therapeutic margins. To address this, we also provide the complete list of predicted compounds, including non-approved agents, to support broader future investigations.

This study has the potential to make a valuable contribution to the field of drug repurposing for cervical cancer. However, the manuscript must be strengthened by addressing the abovementioned limitations, particularly by including more robust biological validation and providing a more nuanced discussion of the confounding factors. By addressing these concerns, the authors can increase the impact and credibility of their findings.

Reviewer 2 Report

Comments and Suggestions for Authors

Interesting study demonstrating potential drug candidates that could be beneficial in the treatment of cervical cancer. The molecular classification into clades α-A7 and α-A9, along with the separate analyses for each group, is also an excellent approach.

One consideration, however, concerns the histologic subtypes of cervical cancer samples included in the study. Carcinomas of the uterine cervix encompass several distinct histologic types, such as:

  • Squamous cell carcinoma, HPV-associated

  • Squamous cell carcinoma, HPV-independent

  • Adenocarcinoma, HPV-associated

  • Adenocarcinoma, HPV-independent

  • Carcinosarcoma of the uterine cervix

  • Adenosquamous and mucoepidermoid carcinomas of the uterine cervix

  • Adenoid basal carcinoma of the uterine cervix

  • Carcinoma of the uterine cervix, unclassifiable

It might be valuable to specify, if possible, which histologic subtypes of carcinoma were included in the analysis, as this could further clarify the molecular and therapeutic implications of the findings.

Comments on the Quality of English Language

line 34 in both 

line 45 after (4,5) comma is missing and Besides is unnecessary

line 59 maybe influences instead of modulates

line 119 within is not needed

line 120 according with is mistake, according to is better

line 219 Strongly should be lowercase

line 251 notably the . Sentence is unfinished.

line 263 related CC.In ... space is missing

Author Response

Interesting study demonstrating potential drug candidates that could be beneficial in the treatment of cervical cancer. The molecular classification into clades α-A7 and α-A9, along with the separate analyses for each group, is also an excellent approach.

One consideration, however, concerns the histologic subtypes of cervical cancer samples included in the study. Carcinomas of the uterine cervix encompass several distinct histologic types, such as:

Squamous cell carcinoma, HPV-associated

Squamous cell carcinoma, HPV-independent

Adenocarcinoma, HPV-associated

Adenocarcinoma, HPV-independent

Carcinosarcoma of the uterine cervix

Adenosquamous and mucoepidermoid carcinomas of the uterine cervix

Adenoid basal carcinoma of the uterine cervix

Carcinoma of the uterine cervix, unclassifiable

It might be valuable to specify, if possible, which histologic subtypes of carcinoma were included in the analysis, as this could further clarify the molecular and therapeutic implications of the findings.

We thank the reviewer for this valuable comment. In the revised version, we clarified the histologic composition of the analyzed samples. As described in the Methods (lines 439–441) and detailed in Supplementary Tables S1–S3, our cohort included the major histologic subtypes reported in the analyzed dataset. We also performed statistical comparisons confirming that the distribution of histologic types was similar across HPV clades, supporting the comparability of the analyzed groups.